# The impact of errors in medical certification on the accuracy of the underlying cause of death

U. S. H. Gamage[1☯], Tim Adair[1☯]*, Lene Mikkelsen[1☯], Pasyodun Koralage Buddhika Mahesh[1☯], John Hart[1‡], Hafiz Chowdhury[1‡], Hang Li[1‡], Rohina Joshi[1,2,3‡], W. M. C. K. Senevirathna[4‡], H. D. N. L. Fernando[4‡], Deirdre McLaughlin[1‡], Alan D. Lopez[5☯]

1 Melbourne School of Population and Global Health, University of Melbourne, Carlton, Victoria, Australia, 2 The George Institute for Global Health, UNSW, Sydney, New South Wales, Australia, 3 The George Institute for Global Health, New Delhi, India, 4 National Institute of Health Sciences, Kalutara, Western Province, Sri Lanka, 5 Institute for Health Metrics and Evaluation (IHME), University of Washington, Seattle, Washington, United States of America

☯ These authors contributed equally to this work.
‡ These authors also contributed equally to this work.
* timothy.adair@unimelb.edu.au

**Data Availability Statement:** All relevant data are within the manuscript and its Supporting information files.

## Abstract

### Background

Correct certification of cause of death by physicians (i.e. completing the medical certificate of cause of death or MCCOD) and correct coding according to International Classification of Diseases (ICD) rules are essential to produce quality mortality statistics to inform health policy. Despite clear guidelines, errors in medical certification are common. This study objectively measures the impact of different medical certification errors upon the selection of the underlying cause of death.

### Methods

A sample of 1592 error-free MCCODs were selected from the 2017 United States multiple cause of death data. The ten most common types of errors in completing the MCCOD (according to published studies) were individually simulated on the error-free MCCODs. After each simulation, the MCCODs were coded using Iris automated mortality coding software. Chance-corrected concordance (CCC) was used to measure the impact of certification errors on the underlying cause of death. Weights for each error type and Socio-demographic Index (SDI) group (representing different mortality conditions) were calculated from the CCC and categorised (*very high*, *high*, *medium* and *low*) to describe their effect on cause of death accuracy.

### Findings

The only *very high impact* error type was *reporting an ill-defined condition as the underlying cause of death*. *High impact* errors were found to be *reporting competing causes in Part 1 [of the death certificate]* and *illegibility*, with *medium impact* errors being *reporting underlying*

**Funding:** This study was funded under an award from Bloomberg Philanthropies and the Australian Department of Foreign Affairs and Trade to the University of Melbourne to support the Data for Health Initiative. The funders had no role in study design, data collection and analysis, decision to publish, or preparation of the manuscript.

**Competing interests:** The authors have declared that no competing interests exist.

*cause in Part 2 [of the death certificate]*, *incorrect or absent time intervals* and *reporting contributory causes in Part 1*, and *low impact* errors comprising *multiple causes per line* and *incorrect sequence*. There was only small difference in error importance between SDI groups.

## Conclusions

Reporting an ill-defined condition as the underlying cause of death can seriously affect the coding outcome, while other certification errors were mitigated through the correct application of mortality coding rules. Training of physicians in not reporting ill-defined conditions on the MCCOD and mortality coders in correct coding practices and using Iris should be important components of national strategies to improve cause of death data quality.

## Introduction

Accurate cause of death statistics are a fundamental component of the evidence base to inform population health policy. They are dependent upon deaths that are correctly certified by a qualified medical practitioner using the World Health Organization's (WHO) International Form of Medical Certificate of Cause of Death (MCCOD), and correctly coded by a trained coder adhering to the rules of the International Classification of Diseases–Version 10 (ICD-10) (S1 Text) [1]. The WHO-recommended medical certificate is comprised of two parts; Part 1 and Part 2. In Part 1, the certifier reports the logical sequence of events leading to death, including the underlying cause of death (UCOD), defined by the WHO as "the disease or injury which initiated the train of morbid events leading directly to death or the circumstances of the accident or violence which produced the fatal injury" [1]. Part 2 is used to report any other significant conditions that may have contributed to death, but that were not part of the morbid sequence initiated by the UCOD. The MCCOD is the primary source of cause of death statistics for much of the world, and thus the basis of interventions to strengthen health and health information systems [2].

High quality cause of death certification (i.e. completing the MCCOD) is strongly dependent on accurately recording the chain of morbid events leading to death in an acceptable sequence, legibly, and without use of nonstandard abbreviations, symptoms, modes of dying and other ill-defined causes that can make coding very difficult [2, 3]. Despite the very clear rules of ICD-10, however, errors in death certification are common and have been noted worldwide [1, 4–13]. The types of errors range from reporting multiple causes on a line of the MCCOD and using abbreviations, to the selection of ill-defined conditions for the UCOD [2]. The more commonly reported errors in the literature that can affect the correct selection of the cause of death are presented in S2 Text [2, 9, 11, 13–18]. Each of these errors can potentially adversely affect the selection of the correct UCOD, and thus, the policy utility of the resultant cause of death statistics.

Efforts to assess and improve the quality of medical certification, especially the training of physicians in accurate certification, should be informed by an evidence-based understanding of the relative importance of medical certification errors on cause of death statistics. Many studies have categorised certification errors according to their perceived potential for affecting the UCOD selection. A common classification method used by many researchers is to subjectively classify errors as being either major or minor, in terms of their perceived impact on diagnostic and coding accuracy [2, 5, 11, 12, 19]. Major errors usually include: (a) reporting multiple causes per line; (b) illegible handwriting; (c) incorrect causal sequence; (d) reporting

ill-defined conditions as the underlying cause of death; (e) missing external cause for deaths due to accidents and violence, and; (f) missing important information about neoplasms [2, 5, 11, 19, 20]. Minor errors typically include: (a) use of abbreviations; (b) missing time intervals, and; (c) leaving blank lines in Part 1 of the medical certificate [2, 5, 20]. Some studies have described the presence of individual error types without categorising them into broad groups [10, 15, 21], while other researchers have studied the impact of errors on specific diseases like cancer, cardiovascular diseases, sudden unexplained deaths etc., or according to their severity [6, 22–24]. Another study assessed the accuracy of death certificate entries compared with postmortem findings, where the researchers graded clinician errors in recording 'Other significant conditions', 'causes of death' and 'manner of death' into categories ranging from no errors to wrong manner of death [13]. However, this kind of error identification and categorization is only possible when the death certificate entries are validated with medical records or autopsy findings; the errors identified in other studies are those specific to completing the MCCOD. None of these studies used empirical evidence to objectively measure the relative importance of the different error types on the accuracy of the resultant cause of death data.

To address this knowledge gap and to improve understanding about the relative importance of medical certification errors, this study uses existing MCCODs to measure the extent to which each type of certification error affects the selection of the UCOD. The results are then used to develop weights to classify the relative importance of each error for countries with different epidemiological profiles. This objective information on the impact of certification errors could subsequently be applied to assess the quality of medical certification in countries and to inform prioritisation of training interventions for physicians.

## Methods

We measured the importance of each error by taking a sample of existing error-free MCCODs and then individually simulating each error type. The error-free MCCODs are those where no medical certification errors were made by the physician in completing the MCCOD. It does not necessarily mean, however, that the UCOD is in fact true, when compared with an autopsy; issues such as diagnostic equipment, physician biases or training, information available for the physician, etc, can affect the accuracy of the UCOD even if there are no certification errors. The resultant underlying cause of death pattern after each error simulation was compared with the pattern from the error-free records using summary validation metrics.

The error-free MCCODs were obtained from the United States' (US) Multiple Cause of Death data file for 2017, which is the only publicly available digitised database of completed MCCODs (i.e. listing every cause reported by the physician in both Part 1 and Part 2 of the death certificate, and identifying the line number and the underlying cause) (S1 Text) [25]. The 2017 dataset comprised 2,820,034 MCCODs which comprised deaths that occurred in all settings: i.e. in hospital, at home, in public places, etc. First, we extracted MCCODs and formed three sample groups that replicated mortality conditions in low, middle and high Socio-demographic Index (SDI) populations. The SDI is an overall index, used in the Global Burden of Disease (GBD) study, of the level of development of a population defined by a composite measure of income per capita, average educational attainment, and fertility rate prevailing in that population [26]. For each sample group, the percentage of deaths in each age group (0–4 years, 5–44 years, 45–64 years, 65–84 years and 85 years and above), sex, and the distribution of deaths across five broad cause categories (Communicable/Maternal/Neonatal/Nutritional; Cardiovascular diseases, Cancers, Other non-communicable diseases; Injuries and accidents) replicated that reported by the GBD study for each SDI population [27]. This process allowed assessment of the impact of each error type in populations with different

epidemiological profiles. For example, in high SDI populations there is typically a higher proportion of deaths at older ages, and from cancer, than in low SDI populations; therefore, if a certain error type has a larger impact on diagnostic accuracy for cancer deaths at older ages, it would be expected that this error might be more important in high rather than low SDI populations.

The sampled MCCODs were then manually screened by trained MCCOD certifiers to remove MCCODs with any type of certification error or those assigned a 'garbage' code (i.e. an unusable code for policy, as defined by the WHO's list of ill-defined causes) as the UCOD [28]. This resulted in 1592 eligible MCCODs, a number of which were allocated to more than one SDI group to reduce the time needed for the manual screening for error-free MCCODs. The final sample comprised 952 deaths representative of high SDI populations, 971 middle SDI populations, and 972 low SDI populations (S1 Table).

Our study assessed the following types of certification errors, identified as being the most common according to published studies (S2 Text) [2, 5, 8, 29, 30]:

1. Incorrect or clinically improbable sequencing of causes in Part 1 of the death certificate

2. Reporting multiple causes on a single line of Part 1

3. Reporting an ill-defined condition as the underlying cause of death in the lowest used line of Part 1

4. Illegible entries (assigned no code or assigned code R99)

5. Incorrect or absent time intervals

6. Reporting competing causes in Part 1

7. Reporting contributory causes in Part 1

8. Reporting underlying causes in Part 2

9. Unspecified neoplasms

10. Poorly defined external cause of death.

In order to identify how each error was simulated, a set of business rules was adopted (S3 Text). These rules were written to replicate, as closely as possible, how these errors are made in practice by certifiers. For the *illegibility* error, given that the error-free database is digitised, we assumed that if a line were illegible the coder would either skip the illegible entry and not assign a code, or assign code R99 (other ill-defined or unknown causes of mortality). Other potential responses by coders include misreading an entry and assigning an incorrect ICD-10 code, or attempting to replace the entry with a plausible alternative based on the other causes reported. These two responses, however, were not feasible options to simulate in our study.

Once MCCODs were simulated with each error, the Iris automated mortality coding software (version 5.7) was applied to select the UCOD [31]. We used the ICD-10 Mortality Tabulation List 1 (103 causes) as the cause list to assess the impact of certification errors on the accuracy of diagnosis [28].

The primary metric used to measure the impact of certification errors was chance-corrected concordance (CCC) [32]. CCC is a measure of the accuracy of individual cause assignment, calculated as:

$$CCC = \frac{\left(\frac{TP}{TP+FN}\right) - \left(\frac{1}{N}\right)}{1 - \left(\frac{1}{N}\right)} \tag{1}$$

where *TP* is true positives (i.e. MCCODs where the underlying cause is the same before and after introduction of the error), *FN* is false negatives (i.e. MCCODs where the underlying cause changed after the introduction of the error), and *N* is the number of MCCODs. Please note that "true positives" is the conventional terminology used for this metric; it does not necessarily mean that the underlying cause of death of an error-free MCCOD is in fact true (when compared to an autopsy).

CCC ranges from 0 (i.e. all diagnoses are wrong) to 1 (i.e. the underlying cause does not change for any MCCODs); the lower the CCC, the greater the adverse impact of the error type on the accuracy of the underlying cause. CCC was calculated for each error type and SDI group (high, middle and low). Weights were calculated based on CCC and for each error type and SDI group, as 1.0 minus CCC; with a higher weighting implying an error type of greater importance.

An aggregate measure of the accuracy of cause-specific mortality fractions (CSMFs) in a population is CSMF accuracy, calculated as [32]:

$$CSMFAccuracy = \frac{\sum \left| CSMF^{true} - CSMF^{pred} \right|}{2 - \left( 1 - Minimum(CSMF^{true}) \right)} \tag{2}$$

where $CSMF^{true}$ is the CSMF from the error-free certificates and $CSMF^{pred}$ is the CSMF from the certificates containing a particular error. Again, we note that the "true" CSMF is simply the conventional terminology used for this metric and does not necessarily mean that the CSMF from error-free MCCODs are in fact true, when compared with an autopsy. CSMF Accuracy also ranges from 0 (i.e. the maximum possible error) to 1 (i.e. the CSMFs based on certification errors had zero impact on correctly specifying the true cause of death distribution). We have used the measure of individual cause assignment (CCC) as a more appropriate metric for assessing the impact of certification errors than CSMF Accuracy, because the impact of certification errors can be masked by "swapping" deaths between causes.

We categorised error types into one of four categories to describe their effect on the accuracy of cause of death data: (1) *Very high impact* (weight 0.40 and higher), (2) *High impact* (weight (0.25<0.40), (3) *Medium impact* (weight 0.10<0.25), and (4) *Low impact* (weight less than 0.10). The category thresholds were chosen to identify key discontinuities in the distribution of weights (Fig 1). The error types *external causes* or *neoplasms* were not categorised because their importance in any given population is dependent on the proportion of deaths that are due to these causes.

We demonstrated the application of the weights for assessing the quality of certification of individual MCCODs. Transformed weights for each error type were calculated as the percentage of the sum of weights across all relevant error types. For deaths not due to neoplasms or external causes, only the weights of the remaining eight error types were included in the calculation. For the calculation of transformed weights for neoplasm deaths, weights for the same eight error types were included, and a weight of 1.0 was applied to the *unspecified neoplasms* error, because the presence of this error means that it is certain that the UCOD of a neoplasm death is incorrect. Similarly, for deaths due to external causes, a weight of 1.0 was applied to the *poorly defined external causes* error. For each individual MCCOD, a composite error score was calculated as the sum of transformed weights of each error type that was present, with the maximum score being 1.0 if all errors were present on the MCCOD, and the minimum score being 0.0 if no errors were present.

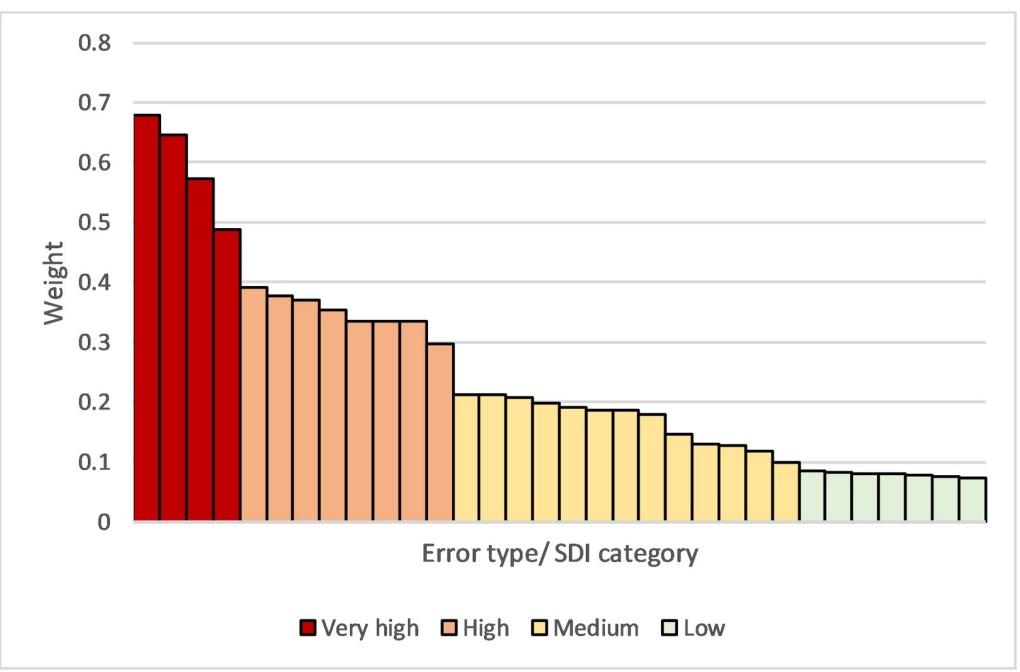

**Fig 1. Error weights of each error type and SDI category.** Very high impact (Red): Weight>0.40. High impact (Orange): Weight 0.25<0.40. Medium impact (Yellow): Weight 0.10<0.25. Low impact (Green): Weight<0.10.

## Results

Table 1 presents the CCC and weights for each error-type and SDI level. For all causes in our sample (high, medium and low SDI populations), the only *very high impact* error type was *ill-defined underlying cause of death*, where over half of all MCCODs had their UCOD changed due to this error (weight 0.573). *High impact* errors across all MCCODs were *reporting*

**Table 1. Chance-corrected concordance (CCC) and weights by error type and SDI level.**

| Error type | All | | SDI level | | | | | |
|---|---|---|---|---|---|---|---|---|
| | | | High | | Middle | | Low | |
| | CCC | Weight | CCC | Weight | CCC | Weight | CCC | Weight |
| Contributory cause in Part 1 | 0.882 | 0.118 | 0.870 | 0.130 | 0.872 | 0.128 | 0.900 | 0.100 |
| Underlying cause in Part 2 | 0.813 | 0.187 | 0.801 | 0.199 | 0.793 | 0.207 | 0.810 | 0.190 |
| Multiple causes per line | 0.926 | 0.074 | 0.916 | 0.084 | 0.920 | 0.080 | 0.925 | 0.075 |
| Incorrect sequence | 0.918 | 0.082 | 0.920 | 0.080 | 0.854 | 0.146 | 0.922 | 0.078 |
| Ill-defined UCOD | 0.427 | 0.573 | 0.321 | 0.679 | 0.354 | 0.646 | 0.511 | 0.489 |
| Competing causes in Part 1 | 0.646 | 0.354 | 0.666 | 0.334 | 0.666 | 0.334 | 0.609 | 0.391 |
| Illegibility (R99 or blank line) | 0.666 | 0.334 | 0.622 | 0.378 | 0.631 | 0.369 | 0.704 | 0.296 |
| Time interval | 0.814 | 0.186 | 0.788 | 0.212 | 0.788 | 0.212 | 0.821 | 0.179 |
| Poorly defined external causes* | 0.941 | 0.059 | 0.950 | 0.050 | 0.921 | 0.079 | 0.925 | 0.075 |
| Unspecified neoplasms* | 0.853 | 0.147 | 0.773 | 0.227 | 0.858 | 0.142 | 0.924 | 0.076 |

Very High Impact (Red): Weight>0.40. High Impact (Orange): Weight 0.25<0.40. Medium Impact (Yellow): Weight 0.10<0.25. Low Impact (Green): Weight<0.10.

* External cause and neoplasms are error types only related to specific types of causes of death, and their importance will depend on the percentage of deaths that are due to each of these causes. Hence, these are not included in the error impact categories.

*competing causes in Part 1* (0.354) and *illegibility* (0.334), *medium impact* errors included *reporting underlying cause in Part 2* (0.187), *time interval* errors (0.186) and *reporting contributory cause in Part 1* (0.118), while *low impact* errors included *incorrect sequence* (0.082) and *multiple causes per line* (0.074). Relatively low weights were calculated for *unspecified neoplasms* (0.147) and *poorly defined external causes* errors (0.059), with these being influenced by the proportion of deaths in the sample that were due to these causes.

The relative importance of the various error types remained relatively consistent across SDI levels. *Ill-defined underlying cause of death* was the most important error type across all SDI levels, being highest in the high SDI category and lowest in the low SDI category. The ill-defined error type also had a lower CCC where only one cause was reported on the death certificate compared with more than one cause (0.454), with this being particularly low in High (0.138) and Middle SDI (0.200) levels (S2 Table). The other notable difference across SDI categories was that the *incorrect sequence* error weight in middle SDI populations was almost double that of high and low SDI populations, and hence was in the *medium* rather than *low impact* category. The *unspecified neoplasms* error was largest in high SDI populations, largely because of the typically higher proportion of neoplasm deaths in these populations. A similar observation was made for *poorly defined external causes* in middle and low SDI populations.

As shown in S3 Table, the weights based on use of CSMF Accuracy as the summary metric of impact were lower than measured using CCC, but the relative importance of the error types was largely consistent across the two impact measures. Of note, *reporting competing causes in Part 1* was only the third most important error according to CSMF accuracy, but was ranked second when CCC was used. *Incorrect sequence* was the least important error according to CSMF Accuracy rather than the second least important according to CCC, with its importance in middle SDI populations declining. These are very minor differences, however, and suggest that the weights are relatively robust across different impact metrics.

Transformed weights were calculated by expressing the weights as a percentage of the sum of all weights. Table 2 presents the transformed weights for each error type and SDI level. For deaths due to neoplasms or external causes, the transformed weights (transformed weights 2) for all other error types were lower than for other causes (transformed weights 1) because all transformed weights must equal 1.0. As an example, if a MCCOD which specified a cardiovascular disease as the cause of death in a high SDI population had *incorrect sequence*

**Table 2. Weights and transformed weights by error type and SDI level.**

| Error type | All | | | High SDI | | | Middle SDI | | | Low SDI | | |
|---|---|---|---|---|---|---|---|---|---|---|---|---|
| | Weight | T/f weight 1 | T/f weight 2 | Weight | T/f weight 1 | T/f weight 2 | Weight | T/f weight 1 | T/f weight 2 | Weight | T/f weight 1 | T/f weight 2 |
| Contributory causes in Part 1 | 0.118 | 0.062 | 0.041 | 0.130 | 0.062 | 0.042 | 0.128 | 0.060 | 0.041 | 0.100 | 0.056 | 0.036 |
| Underlying cause in Part 2 | 0.187 | 0.098 | 0.064 | 0.199 | 0.095 | 0.064 | 0.207 | 0.098 | 0.066 | 0.190 | 0.106 | 0.068 |
| Multiple causes per line | 0.074 | 0.039 | 0.025 | 0.084 | 0.040 | 0.027 | 0.080 | 0.038 | 0.026 | 0.075 | 0.042 | 0.027 |
| Incorrect sequence | 0.082 | 0.043 | 0.028 | 0.080 | 0.038 | 0.026 | 0.146 | 0.069 | 0.047 | 0.078 | 0.043 | 0.028 |
| Ill-defined UCOD | 0.573 | 0.300 | 0.197 | 0.679 | 0.324 | 0.219 | 0.646 | 0.304 | 0.207 | 0.489 | 0.272 | 0.175 |
| Competing causes in Part 1 | 0.354 | 0.186 | 0.122 | 0.334 | 0.159 | 0.108 | 0.334 | 0.157 | 0.107 | 0.391 | 0.217 | 0.140 |
| Illegibility (R99 or blank line) | 0.334 | 0.175 | 0.115 | 0.378 | 0.180 | 0.122 | 0.369 | 0.174 | 0.118 | 0.296 | 0.165 | 0.106 |
| Time interval | 0.186 | 0.097 | 0.064 | 0.212 | 0.101 | 0.068 | 0.212 | 0.100 | 0.068 | 0.179 | 0.100 | 0.064 |
| Poorly defined external causes / Unspecified neoplasms | 1.000 | - | 0.344 | 1.000 | - | 0.323 | 1.000 | - | 0.320 | 1.000 | - | 0.357 |

T/f weight 1: Transformed weights for MCCODs of deaths not due to neoplasms or external causes. T/f weight 2: Transformed weights for MCCODs of deaths due to either neoplasms or external causes.

(Transformed weight 0.038), *reporting contributory causes in Part 1* (0.062) and *multiple causes per line* (0.040) as errors, then the composite score of that MCCOD would be 0.140. The same errors on an MCCOD for an external cause of death would result in a composite score of 0.095 (0.026 + 0.042 + 0.027).

## Discussion

We believe that this is the first study to use empirical evidence to measure the relative impact of various common medical certification errors on the UCOD, generating important information to inform efforts to improve medical certification. Common certification errors were identified by searching the published literature [2, 6–9, 11, 13–18, 24, 30], business rules were developed to reflect how these errors occur in practice, and each error was simulated separately on 1592 error free death certificates and then coded using Iris automated coding software. Weights were calculated for different SDI levels (low, middle, high) to measure the impact of the errors for countries at different stages of development and to enable the monitoring of trends in the quality of medical certification.

Overall, the impact of each error type upon the selection of the UCOD appears to be less than what was expected. This was particularly evident with errors such as *incorrect sequence* and *multiple causes per line*, which were previously considered to be 'major errors' by researchers [2, 5, 11, 19]. This could in part be due to the mitigating effect of cause of death coding rules. While in most cases the UCOD is that identified to be the condition initiating the causal sequence, sometimes a condition other than that which initiated the morbid sequence leading to death is selected as the UCOD according to international coding rules that consider epidemiological and other public health factors in the UCOD selection. The ICD-10 mortality coding rules are designed in such a way that they can partially mitigate the impact of possible certification errors on the selection of the UCOD [28].

The one error type that was consistently identified as having a *very high* impact was the *reporting of an ill-defined condition as the underlying cause of death*, most notably in high SDI populations and where only one cause was reported on the death certificate. This finding is to be expected given that an ill-defined underlying cause of death, by definition, should lead to a misdiagnosis of the true cause of death; something we observed for more than half of the deaths. *Illegibility* and *reporting competing causes in Part 1 of the death certificate* were the next most important errors (affecting the underlying cause in approximately 30 to 40% of MCCODs), followed by *time interval* errors and *reporting contributory causes in Part 1*. *Multiple causes on the same line* and *specifying an incorrect causal sequence* generally had a *low impact* or, for the latter in middle SDI countries, *medium impact* on the selection of the true UCOD, yet in the published literature, 'incorrect causal sequencing' and 'multiple causes in a single line' have been considered as major errors [2, 19]. This empirical finding is perhaps surprising, and demonstrates the beneficial impact of coding rules SP4 and SP5 [28] on correctly specifying the underlying cause. Rule SP4 allows the coder to search for another possible sequence ending in the first mentioned condition (terminal condition) in part 1 and select the originating cause of that sequence as the tentative starting point. Further, when there is no sequence ending with the terminal condition, rule SP5 allows the coder to select the terminal condition also as the tentative starting point. For most error types, there was only a small difference in impact across SDI levels. This demonstrates that the impact of error types is largely invariant to differences in age, sex and cause composition of deaths in a population, and their relative weight is consistent irrespective of the country to which they are applied.

For two errors specific to particular causes of death, neoplasms and external causes, the results need to be interpreted in isolation from other errors because their importance is

dependent on the proportion of MCCODs with each of these causes. When neoplasms are insufficiently specified in terms of site, morphology, and behaviour, the resultant UCOD is always incorrect. The *unspecified neoplasms* error was largest in high SDI populations because the highest proportion of neoplasm deaths occurs in these populations. Similarly, the *poorly defined external causes* error was most prevalent in middle and low SDI populations, where the highest proportion of deaths due to this cause occur.

The comparative magnitude of the transformed weights derived in this study has significant implications for guiding intervention efforts to improve the quality of medical certification practices. By measuring the relative importance of each certification type and categorising them into groups based on their impact, they inform prioritisation of the errors medical certification training should focus on to have the maximum impact on resultant cause of death quality. The weights also allow development of a composite index of certification quality which can be used to monitor trends and differentials in overall certification practices from a database of MCCODs. This index is a better measure of cause of death certification quality than the indices based on judgmental error weights, because it is based on empirical evidence about the comparative effects of individual errors on the selection of the UCOD.

One limitation of this study is that Iris automated coding software, although ensuring consistent application of mortality coding rules after each error simulation, is not widely used in many low- and middle-income countries [33]. Instead, coding is performed manually, a process which may be subject to incorrect or inconsistent application of coding rules. With the correct application of coding rules mitigating, to some extent, the importance of errors, it may be that in many settings these errors consequently have a greater impact on the UCOD selection than what we have shown in this study. Additionally, the relatively high quality of the error-free certificates obtained from the US database, which is of higher quality than data in many low- and middle-income countries, may also have played a part in the lower than expected impact of errors. Using the *incorrect sequence* error as an example, for a death certificate from the US it is very likely that the correct UCOD will be reported by the doctor somewhere on the certificate, even if the sequence is incorrect. Therefore, even if causes are jumbled within a death certificate, the correct UCOD will likely still be listed somewhere on the certificate, making it possible for Iris to select the correct UCOD. In a setting where the quality of medical certification is lower and the correct UCOD may not be mentioned anywhere on the certificate, this may not occur. Hence, the magnitude of the impact of each error on the accuracy of cause of death statistics in low- and middle-income countries may be higher than what is presented in this study. However, it is likely that the results will still be relevant for countries with different levels of certification quality because we have attempted to measure impact for epidemiological profiles of low, middle and high SDI populations.

Another potential limitation of our study was that the errors were introduced individually to measure the impact of each separately, rather than assessing the combined effects of multiple errors, which is likely to occur in practice [2, 24]. Assessing the diagnostic impact of clusters of errors would however require a much larger data set and would involve vast possible combinations of two or more of the ten errors. For example, assessment of the ill-defined error may need to be conducted with one of each of the other errors, different combinations of each of the other errors, up to all eight other errors; this would make disentangling the relative importance of each error a major methodological challenge that is beyond the scope of this study. Additionally, combining multiple errors would need to be done with some empirical understanding of the composition of error clusters, which are likely to also differ by cause. On the other hand, individual assessment of errors, as we have done, provides comparative information on the relative importance of various errors and in a more robust way because it is not subjected to the impact of one error being reduced by the presence of another error.

We also note that the use of a digitised database without time intervals reduced our ability to assess *illegibility* and *time intervals* errors. Nonetheless, we were able to simulate these errors to closely mimic their appearance in practice. Further, while 'use of abbreviations' is another commonly identified error type, it was not simulated in this study due to the difficulty in predicting coders' interpretations of non-standard abbreviations used by certifying physicians. We also measured weights based on their importance using samples of MCCODs with the age-sex and cause pattern of large populations. For analyses of MCCOD quality in settings with different cause of death profiles, such as specialist (e.g. cancer) hospitals, some transformed weights may reflect higher or lower values than their actual impact (e.g. the unspecified neoplasms error would have a lower score than its true impact). Finally, the UCOD of the error-free MCCODs may not be the "true" UCOD, when measured against a gold standard. However, this should not affect the relative impact of each type of certification error on the cause of death statistics.

Despite these limitations, the empirical evidence generated on the comparative importance of different certification errors on cause of death accuracy will have, we believe, important implications for guiding efforts to improve diagnostic accuracy. In particular, the finding that *reporting an ill-defined condition as the underlying cause of deat*h substantially and consistently has a high impact on the accuracy of cause of death statistics, particularly in high SDI populations, suggests clear priorities for physician training programs in all countries. In addition, medical certification quality assessment tools can now use these evidence-based findings to objectively assess the quality of medical certification training and practice.

## Conclusion

Different certification errors have a variable impact on the selection of the underlying cause of death. The greatest impact results from the reporting of ill-defined conditions as the underlying cause of death. Illegibility of the entries and reporting competing causes in Part 1 of the death certificate have a medium impact on the selection of the underlying cause of death. Other errors, including reporting multiple causes on a single line, incorrect sequencing of causes, reporting contributory causes in Part 1, reporting underlying causes in Part 2, insufficiently specified neoplasms, and insufficiently specified external causes do not appear to have much impact on the selection of the underlying cause of death. The impact of certification errors on the selection of underlying cause of death does not vary much between low, middle and high SDI populations. Although we were not able to use a database from a low- or middle-income country, our findings are generalizable to such countries because we sample MCCODs with the cause, age and sex profile of low, middle and high SDI countries. Correct application of mortality coding rules can substantially mitigate the effects of some certification errors. Training mortality coders in correct coding practices and using automated coding software should, therefore, be an important component of national strategies to improve the quality of cause of death data, along with training physicians in correct certification practices.

## Supporting information

**S1 Text. International form of the Medical Certificate of Cause of Death (MCCOD).**
(DOCX)

**S2 Text. Certification errors.**
(DOCX)

**S3 Text. Business rules used.**
(DOCX)

**S1 Table. Age, sex and cause distribution of MCCODs in final sample (%), low, middle, high SDI countries and all.**
(DOCX)

**S2 Table. Chance-corrected concordance for ill-defined UCOD error, by number of causes reported on death certificate and SDI level.**
(DOCX)

**S3 Table. CSMF accuracy and weights by error type and SDI level.**
(DOCX)

**S1 Data.**
(CSV)

## Acknowledgments

The authors wish to acknowledge the contribution of Sara Hudson for her assistance with the editing and review of the final format of the document.

## Author Contributions

**Conceptualization:** U. S. H. Gamage, Lene Mikkelsen, Deirdre McLaughlin, Alan D. Lopez.

**Data curation:** U. S. H. Gamage, Tim Adair, Pasyodun Koralage Buddhika Mahesh, W. M. C. K. Senevirathna, H. D. N. L. Fernando.

**Formal analysis:** U. S. H. Gamage, Tim Adair, Pasyodun Koralage Buddhika Mahesh.

**Funding acquisition:** Alan D. Lopez.

**Investigation:** W. M. C. K. Senevirathna, H. D. N. L. Fernando.

**Methodology:** U. S. H. Gamage, Tim Adair, Pasyodun Koralage Buddhika Mahesh, John Hart, Hafiz Chowdhury, Hang Li, Rohina Joshi, Alan D. Lopez.

**Supervision:** Deirdre McLaughlin, Alan D. Lopez.

**Validation:** U. S. H. Gamage, Tim Adair, Lene Mikkelsen, Pasyodun Koralage Buddhika Mahesh, John Hart, Hafiz Chowdhury, Hang Li, Rohina Joshi.

**Writing – original draft:** U. S. H. Gamage, Tim Adair.

**Writing – review & editing:** U. S. H. Gamage, Tim Adair, Lene Mikkelsen, Pasyodun Koralage Buddhika Mahesh, John Hart, Hafiz Chowdhury, Hang Li, Rohina Joshi, W. M. C. K. Senevirathna, H. D. N. L. Fernando, Deirdre McLaughlin, Alan D. Lopez.

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
