## [Decision Letter · Decision Letter 0]

29 Jun 2021

PONE-D-21-14961

The impact of errors in medical certification on the diagnostic accuracy of the underlying cause of death

PLOS ONE

Dear Dr. Adair,

Thank you for submitting your manuscript to PLOS ONE. After careful consideration, we feel that it has merit but does not fully meet PLOS ONE’s publication criteria as it currently stands. Therefore, we invite you to submit a revised version of the manuscript that addresses the points raised during the review process.

Please submit your revised manuscript by 19-July-2021. Please include the following items when submitting your revised manuscript:

A rebuttal letter that responds to each point raised by the academic editor and reviewer(s). You should upload this letter as a separate file labeled 'Response to Reviewers'.A marked-up copy of your manuscript that highlights changes made to the original version. You should upload this as a separate file labeled 'Revised Manuscript with Track Changes'.An unmarked version of your revised paper without tracked changes. You should upload this as a separate file labeled 'Manuscript'

We look forward to receiving your revised manuscript.

Kind regards,

Prof. Ritesh G. Menezes, M.B.B.S., M.D., Diplomate N.B.

Academic Editor

PLOS ONE

Journal Requirements:

Reviewers' comments:

Reviewer's Responses to Questions

**Comments to the Author**

1. Is the manuscript technically sound, and do the data support the conclusions?

Reviewer #1: No

Reviewer #2: Yes

Reviewer #3: Yes

2. Has the statistical analysis been performed appropriately and rigorously? 

Reviewer #1: No

Reviewer #2: Yes

Reviewer #3: Yes

3. Have the authors made all data underlying the findings in their manuscript fully available?

Reviewer #1: Yes

Reviewer #2: Yes

Reviewer #3: Yes

4. Is the manuscript presented in an intelligible fashion and written in standard English?

Reviewer #1: No

Reviewer #2: Yes

Reviewer #3: Yes

5. Review Comments to the Author

Reviewer #1: This is an extremely important topic.

Rejection causes:

Major:

1. Multiple recognized (recent and high impact) Major and Minor diagnosis errors systems have been published but ignored.

2. The basis of the study is the impact of the errors. This can NOT be calculated if the true cause is UNKNOWN. If there is a focus on reading recent (and old) publications the authors will understand that this is the reason data and statistics from current death certificates can NOT be used. Regardless what sophisticated statistics are used with "garbage" data, it will still result in erroneous results, discussion and conclusion (garbage in, garbage out).

Minor:

1, Grammar and typo errors (multiple). Many that would be highlighted and corrected by word or google documents

2. Old references (except those by authors. Self referencing (while excluding others) gives a false impression of importance without recognizing others

3. Repetition of reference 14 and 15 (30 and 32)

Reviewer #2: This is a very interesting paper dealing with an essential aspect of public health monitoring: the impact of errors in medical certification on the diagnostic accuracy of the underlyng cause of death. This is the first study I read about measuring the relative impact of various common medical certification errors on the UCOD and that gives important results to improve medical certification.

Following are my comments:

1. The error-free MCCODs were obtained from the United States’ (US) Multiple Cause of Death data file for 2017 and after that a sample of 1592 MCCODs were used. It would be important to know how many MCCODs were available in the US data file.

2. About the reporting of an ill-defined condition as the UCOD error, I would like to know if there was more impact in the MCCODs that had only one line written in part 1 or if there were other ill-defined cause of death in all the other lines of part 1.

3. About the illegibility error, you mentioned that if there were an illegible line, the coder would skip the illegible entry and not assign a code, how do you difference that from a blank space left by the medical certifier.

4. It would have been important to analyse the use of abbreviations, at least for the countries using Iris coding.

Reviewer #3: 1.A well-written, neatly structured manuscript

2.The manuscript is technically sound and the data does support the conclusions

3. Statistical analysis and explanation of results is clear

4. Conclusion needs to be redone to incorporate lot more information

6. PLOS authors have the option to publish the peer review history of their article (what does this mean?). If published, this will include your full peer review and any attached files.

Reviewer #1: No

Reviewer #2: **Yes: **Janet Miki

Reviewer #3: **Yes: **Dr Kavya Rangaswamy

---

## [Author Response · Author response to Decision Letter 0]

15 Jul 2021

Responses to reviewers’ comments 

Review Comments to the Author

We thank the Editor for providing us the opportunity to respond to the reviewers’ valuable comments. Please find our responses below in red. Line numbers refer to the clean version of the manuscript.

Reviewer #1: This is an extremely important topic.

We thank the reviewer for their comments. Please find our responses below. Line numbers refer to the tracked version of the manuscript.

We would like to clarify that our study assesses the impact on the accuracy of cause of death statistics of errors made by physicians in completing the Medical Certificate of Cause of Death, as according to the WHO standard guidelines in certification. We did not specifically assess the diagnostic accuracy of the diseases and conditions reported on the certificate against an autopsy. Hence, the “error-free” death certificates are those where the physician has made no errors in completing the death certificate, based on the WHO standard guidelines in certification. It is likely that many of these “error-free” death certificates would not have the “true” cause of death, as assessed when compared with an autopsy, for various reasons (e.g. diagnostic limitations, biases of the physician, etc). Our assessment of the accuracy of cause of death statistics compares, for each death in the database, the underlying cause of death (after the application of ICD coding rules) where there is each type of certification error versus the underlying cause of death where there is no error on the death certificate.

To reflect this, we have amended the title of the manuscript (removing the term “diagnostic”) and also clarified the specific nature of the study and methods used in these places:

- Abstract Lines 26-27 and Line 33-34

- Introduction Line 69

- Methods Lines 111-115, 171-173, 185-87

- Discussion Lines 359-61

Rejection causes:

Major:

1. Multiple recognized (recent and high impact) Major and Minor diagnosis errors systems have been published but ignored.

We based our categorisation of certification error types on those identified as the most common according to published studies. The reviewer has stated that there are other diagnosis errors systems, however the errors that we are focusing on are errors in completing the death certificate according to WHO guidelines and how these impact on the accuracy of causes of death (after the application of ICD coding rules), rather than errors specifically related to diagnostic accuracy of the diseases and conditions reported on the certificate. We note that a recent article by Schuppener et al. (Death certification: errors and interventions. Clinical Medicine & Research, 2020;18(1):21-6) identified “Wrong COD” as an error type (we have added a description of this study in Lines 92-99). However, this kind of error identification and categorization is only possible when the death certificate entries are validated with medical records or autopsy findings, and is not applicable to assessment of errors in completing the MCCOD.

2. The basis of the study is the impact of the errors. This can NOT be calculated if the true cause is UNKNOWN. If there is a focus on reading recent (and old) publications the authors will understand that this is the reason data and statistics from current death certificates can NOT be used. Regardless what sophisticated statistics are used with "garbage" data, it will still result in erroneous results, discussion and conclusion (garbage in, garbage out).

As mentioned above, our aim was to measure the impact of each different type of error in completing the death certificate (according to WHO guidelines) on the resultant cause of death (after application of ICD coding rules). We do this by comparing the cause of death of the death certificates with an error against those without an error, using the metric of chance-corrected concordance. However, we do recognise that we do not know the true cause of each death, which may be impacted by diagnostic equipment, physician biases or training, information available for the physician, etc. We don’t necessarily believe the error-free certificates have the ‘true’ cause of death (as compared with an autopsy) but the cause of death where none of certification errors are present. We have clarified this issue in Lines 111-115 and 359-61.

We do not agree however that data and statistics from current death certificates cannot be used. In fact, they are used widely, including by the WHO, Global Burden of Disease Study and numerous national statistics offices. There are established methods to assess the extent and severity of garbage codes and reallocate these to non-garbage causes (see Naghavi M et al. Improving the quality of cause of death data for public health policy: are all 'garbage' codes equally problematic? BMC Med. 2020;18(1):55, and Naghavi M, et al. Algorithms for enhancing public health utility of national causes-of-death data. Popul Health Metr. 2010;8:9). We also recognise that it is not possible to conduct an autopsy on all deaths, so physician-completed death certificates are the primary source of information on causes of death. We also recognise that there should be continued efforts to improve training of physicians in death certification is important to ensure that the data are as useful for policy as possible.

Minor:

1, Grammar and typo errors (multiple). Many that would be highlighted and corrected by word or google documents

We have amended grammar and typo errors in multiple places.

2. Old references (except those by authors. Self referencing (while excluding others) gives a false impression of importance without recognizing others

We didn’t intend to create a false impression of importance. We have now added more recent references of Schuppener et al, 2020 (reference #13, described in lines 92-97) and Brooks & Reed, 2015 (reference #12, Lines 73, 84). We note that other references we have used are relatively recent: McGiven 2017, Flippatos 2016, Miki 2018, Crandll 2020, Azim 2014.

3. Repetition of reference 14 and 15 (30 and 32)

Thank you for identifying these issues - we have deleted these duplicate references.

Reviewer #2: This is a very interesting paper dealing with an essential aspect of public health monitoring: the impact of errors in medical certification on the diagnostic accuracy of the underlyng cause of death. This is the first study I read about measuring the relative impact of various common medical certification errors on the UCOD and that gives important results to improve medical certification.

We thank the reviewer for providing valuable comments on the manuscript. Please find our responses below. Line numbers refer to the tracked version of the manuscript.

Following are my comments:

1. The error-free MCCODs were obtained from the United States’ (US) Multiple Cause of Death data file for 2017 and after that a sample of 1592 MCCODs were used. It would be important to know how many MCCODs were available in the US data file.

There were a total of 2,820,034 MCCODs in the UCOD data file in 2017 – we have now included this figure in Line 121. We selected a sample of MCCODs because each MCCOD needed to be individually screened for errors by trained MCCOD certifiers so that they could select the error-free certificates.

2. About the reporting of an ill-defined condition as the UCOD error, I would like to know if there was more impact in the MCCODs that had only one line written in part 1 or if there were other ill-defined cause of death in all the other lines of part 1.

The reviewer raises a good point. We conducted a further analysis of whether there was greater impact on MCCODs of the ill-defined error with only one cause written in Part 1 of the death certificate compared with more than one cause. The results shown in the table below show that the chance-corrected concordance for the ill-defined error with only one cause reported is 0.353 compared with 0.454 for more than one error. Notably, the chance-corrected concordance for the ill-defined error with only one cause reported is particularly low in high SDI (0.138) and middle SDI (0.200) countries, but with less of an impact in low SDI countries where the problem of only one cause being reported may be more pronounced.

We have included this table in S2 Table and briefly described the results in Lines 229-231 and 286-287.

S2 Table. Chance-Corrected Concordance for ill-defined UCOD error, by number of causes reported on death certificate and SDI level

 SDI level

Number of causes reported All High Middle Low

Ill-defined UCOD: 1 cause 0.353 0.138 0.200 0.457

Ill-defined UCOD: >1 cause 0.454 0.385 0.401 0.532

3. About the illegibility error, you mentioned that if there were an illegible line, the coder would skip the illegible entry and not assign a code, how do you difference that from a blank space left by the medical certifier.

In our simulation of the illegibility error, one assumption we made was that the coder would skip the illegible entry and not assign a code. In practice, this may be the same as if the certifier did leave a blank space. Leaving a blank space will generally cause a lesser impact on coding and the selection of the underlying cause than the removal of a condition from the certificate due to illegibility. Blank spaces are usually ignored during the coding process, but the inability to code due to illegibility means a removal of a condition which had been originally reported by the certifier. The latter causes a higher impact on the selection of underlying cause than blank spaces. 

4. It would have been important to analyse the use of abbreviations, at least for the countries using Iris coding.

We agree that it would have been valuable to analyse the use of abbreviations, however we were unable to do so because we were using a digitised database and we could not identify a plausible means of simulating this error. We have mentioned this issue already in Lines 353-355. However, any abbreviation can be included into the Iris dictionary using standardization tables, which can lessen the impact of abbreviations on cause of death statistics.

Reviewer #3: 1.A well-written, neatly structured manuscript

2.The manuscript is technically sound and the data does support the conclusions

3. Statistical analysis and explanation of results is clear

4. Conclusion needs to be redone to incorporate lot more information

We thank the reviewer for their valuable comments.

1. Since the study uses error-free MCCODS from US database, generalizing the results across the globe (primarily in medium and low income countries) would be wrong. In fact, the study states that the impact of errors on selection of UCOD is lesser than expected. if the sample was MCCODs collected from diverse countries the results would be different altogether. The authors need to emphasize on this point lot more both in the discussion and conclusion

We would like to clarify how we sought to generalise the results from our study to middle and low income (Socio-Demographic Index or SDI) countries. In the study, we selected samples of MCCODs that represent the age, sex and cause profile of low, middle and high SDI countries, to provide results that estimate how each error type would impact cause of death accuracy across the different epidemiological profiles present in the world. For example, the Low SDI sample has a higher proportion of deaths from infectious diseases compared with the High SDI sample (see S1 Table). Unfortunately, we were unable to use a similar database from a low- or middle-income country. In the discussion, we have already stated in Lines 325-327 that “Additionally, the relatively high quality of the error-free certificates, obtained from a US database which is of higher quality than databases in many low- and middle-income countries, may also have played a part in the lower than expected impact of errors.” We have however now added in the Discussion in Lines 332-334 that the magnitude of the results may be different if we were able to use data from a low- or middle-income country. We have also mentioned this overall limitation in the Conclusion (Lines 381-383).

3. Though the list of errors in death certification are similar across countries, their impact on diagnostic accuracy of UCOD is varied especially in medium and low income countries. Emphasis on this differentiation is not evident in the manuscript.

We primarily based our interpretation of our results in terms of their differential impact on high, middle and low SDI countries from Table 1. This shows that there are considerable differences in the chance-corrected concordance of the ill-defined error, especially where only one cause is reported in the MCCOD (see new S2 Table and Lines 229-231). Other clear differences are found for the unspecified neoplasms error, which is already reported in Lines 234-235. However, for none of the other errors is there a clear difference in the Chance-corrected concordance or Weight (and impact category) between any of the SDI levels, so we believe that it is not necessary to report these.

4. As mentioned in the limitation of the study, the impact of combined errors is not studied. The author needs to articulate much further the advantages of studying individual errors 

The reviewer makes a good point. As we have mentioned in Lines 338-349, the main advantages of analysing individual errors is that it allows individual measurement of the impact of each error without being biased by the presence of other errors, that we do not need to make assumptions of the composition or combination of errors on MCCODs and that it does not require using a vast number of possible combination of causes. We have now expanded on the last of these issues in Lines 340-345, specifically that: 

“Assessing the diagnostic impact of clusters of errors would however require a much larger data set and would involve vast possible combinations of two or more of the nine errors. For example, assessment of the ill-defined error may need to be conducted with one of each of the other errors, different combinations of each of the other errors, up to all eight other errors; this would make disentangling the relative importance of each error a major methodological challenge that is beyond the scope of this study.”

5. Conclusion needs to be redone to incorporate lot more information

We are unsure which specific information should be added to the Conclusion, keeping in mind that it should only be relatively brief. However, in the Conclusion we have now included information on the overall limitation of the use of the selection of samples of MCCODs from the US database that represent the age, sex and cause profile of low, middle and high SDI countries due to no alternate database being available from low- or middle-income countries. 

Do you want your identity to be public for this peer review? For information about this choice, including consent withdrawal, please see our Privacy Policy.

Reviewer #1: No

Reviewer #2: Yes: Janet Miki

Reviewer #3: Yes: Dr Kavya Rangaswamy

---

## [Decision Letter · Decision Letter 1]

7 Oct 2021

PONE-D-21-14961R1

The impact of errors in medical certification on the accuracy of the underlying cause of death

PLOS ONE

Dear Dr. Adair,

Thank you for submitting your manuscript to PLOS ONE. After careful consideration, we feel that it has merit but does not fully meet PLOS ONE’s publication criteria as it currently stands. Therefore, we invite you to submit a revised version of the manuscript that addresses the points raised during the review process.

Please submit your revised manuscript by 14-October-2021. Please include the following items when submitting your revised manuscript:

A 'Response to Reviewers' letter that responds to each point raised by the academic editor and reviewer(s). You should upload this letter as a separate file labeled 'Response to Reviewers'.A marked-up copy of your manuscript that highlights changes made to the original version. You should upload this as a separate file labeled 'Revised Manuscript with Track Changes'.An unmarked version of your revised paper without tracked changes. You should upload this as a separate file labeled 'Manuscript'.

We look forward to receiving your revised manuscript.

Kind regards,

Prof. Ritesh G. Menezes, M.B.B.S., M.D., Diplomate N.B.

Academic Editor

PLOS ONE

Journal Requirements:

Reviewers' comments:

Reviewer's Responses to Questions

**Comments to the Author**

1. If the authors have adequately addressed your comments raised in a previous round of review and you feel that this manuscript is now acceptable for publication, you may indicate that here to bypass the “Comments to the Author” section, enter your conflict of interest statement in the “Confidential to Editor” section, and submit your "Accept" recommendation.

Reviewer #2: All comments have been addressed

Reviewer #4: All comments have been addressed

Reviewer #5: (No Response)

Reviewer #6: All comments have been addressed

2. Is the manuscript technically sound, and do the data support the conclusions?

Reviewer #2: Yes

Reviewer #4: Yes

Reviewer #5: Partly

Reviewer #6: Yes

3. Has the statistical analysis been performed appropriately and rigorously? 

Reviewer #2: Yes

Reviewer #4: Yes

Reviewer #5: Yes

Reviewer #6: Yes

4. Have the authors made all data underlying the findings in their manuscript fully available?

Reviewer #2: Yes

Reviewer #4: Yes

Reviewer #5: Yes

Reviewer #6: Yes

5. Is the manuscript presented in an intelligible fashion and written in standard English?

Reviewer #2: Yes

Reviewer #4: Yes

Reviewer #5: Yes

Reviewer #6: Yes

6. Review Comments to the Author

Reviewer #2: (No Response)

Reviewer #4: (No Response)

Reviewer #5: A well written and novel approach to estimating the extent to which MCCODS errors affect the UCOD. Kindly find herein my comments on the article:

• Line 43-42 : The authors can mention what errors, if any, came under the “Low impact error type” as the paragraph mentions the classification of error types into very high, high, medium, and low.

• Line 49 : Explanation of what Iris stands for, (the automated coding software), would give an uninitiated reader a better understanding as it may be a new term for many young physicians.

• Line 60 : The S1 text reference only shows the guidelines for filling out the WHO MCCOD. However, there should also be a reference for the guidelines for Coding mentioned here as it has been specifically mentioned in lines 59-60.

• Line 72 : Does the “very clear rules” mentioned here refer to lines 70-71? If not, then does it refer to S1 text reference? Or is it mentioned among the other references (4-13)? More clarity is needed on this.

• S2 text : “Reporting multiple causes in a single line of part 1” and “Reporting competing causes in part 1” appear at first glance to be separate entities. So, does that imply that the multiple causes in the single line are not mutually exclusive but equally possible? And the competing causes of part 1 may also be in a single line. Which type of error would it then come under? As the result is essentially wrongly coding 1 of the causes listed as the UCOD. A better explanation/differentiation between the two types of errors would be helpful.

• Line 123 and footnote 1 : Though the explanation for how SDI is calculated is mentioned, since the authors have mentioned that it is used by Global burden of disease study, a reference to a GBD capstone article/ related GBD reference may be included.

• Line 136 : Would the decision to include the MCCODs in multiple SDI groups not affect the results? The rationale for including them in multiple SDI groups needs to be mentioned for better clarity. The authors also need to clarify here whether MCCODS with “unspecified neoplasms” error and “poorly defined external causes” error were also out in multiple SDI groups or not.

• Fig 1 : SDI categories are not labelled.

• Lines 272-273 : Though one of the stated goals was “to monitor trends in the quality of medical certification”, subsequent sections fail to elaborate on this. Though there is the mention of the use of Iris in mainly high SDI countries and the possibility of manual coding causing more errors, there isn’t enough discussion to correlate the study findings and the “quality of medical certification” in other countries.

• Lines 279-281 : More details need to be mentioned regarding situations where epidemiological and other public health factors are considered in the UCOD selection. Since these coding rules are mentioned as the main reason for mitigating errors, the situations where they are applied instead of using the condition initiating the causal sequence needs to be elaborated. And if according to the coding rules, some other condition is selected as the UCOD and not the condition initiating the causal sequence, should it still be considered as an error for the purpose of this study?

• Line 296 : A brief explanation of Coding rules SP4 and SP5 is needed even though a reference for the same is given (24). This is because of the importance of these rules in understanding why the effect of published “major errors” may not be as high as expected.

Reviewer #6: Dear Authors,

The authors have satisfactorily addressed all the queries by the previous reviewers. This article is very relevant in the present scenario of COVID pandemic. The authors are requested to address the below queries .

Abstract:

• Line 32 and 33: The author needs to clarify whether “ 1592- error free reports” selected were “ either the death occurred in the medical institution or non- institutional death”, since WHO has recommended two types of Forms specifically ie FORM 4 and FORM 4A.

Discussion:

• Line 328: The author needs to write the type of version and make of “ Iris automated coding software” and the calibration to avoid errors in coding.

7. PLOS authors have the option to publish the peer review history of their article (what does this mean?). If published, this will include your full peer review and any attached files.

Reviewer #2: **Yes: **Janet Miki

Reviewer #4: **Yes: **Prateek rastogi

Reviewer #5: No

Reviewer #6: **Yes: **Dr Jagadish Rao Padubidri

---

## [Author Response · Author response to Decision Letter 1]

21 Oct 2021

PONE-D-21-14961R1: The impact of errors in medical certification on the accuracy of the underlying cause of death

Responses to reviewers’ comments

We thank the Editor for the opportunity to respond to the reviewers’ comments. Please find below our responses. All line numbers refer to the document with tracked changes.

Reviewer #5: A well written and novel approach to estimating the extent to which MCCODS errors affect the UCOD. Kindly find herein my comments on the article:

We thank the reviewer for their valuable comments. Please find below our responses. All line numbers refer to the document with tracked changes.

• Line 43-42 : The authors can mention what errors, if any, came under the “Low impact error type” as the paragraph mentions the classification of error types into very high, high, medium, and low.

We have now added that the low impact errors were multiple causes per line and incorrect sequence. 

• Line 49 : Explanation of what Iris stands for, (the automated coding software), would give an uninitiated reader a better understanding as it may be a new term for many young physicians.

Iris is an automated mortality coding software – we now state this in Line 35.

• Line 60 : The S1 text reference only shows the guidelines for filling out the WHO MCCOD. However, there should also be a reference for the guidelines for Coding mentioned here as it has been specifically mentioned in lines 59-60.

We have now added Reference 1 (WHO ICD – 10 vol 2) in Line 62.

• Line 72 : Does the “very clear rules” mentioned here refer to lines 70-71? If not, then does it refer to S1 text reference? Or is it mentioned among the other references (4-13)? More clarity is needed on this.

These “very clear rules” are those in WHO ICD – 10 vol 2 (reference #1) – we have now stated this in Lines 74-75.

• S2 text : “Reporting multiple causes in a single line of part 1” and “Reporting competing causes in part 1” appear at first glance to be separate entities. So, does that imply that the multiple causes in the single line are not mutually exclusive but equally possible? And the competing causes of part 1 may also be in a single line. Which type of error would it then come under? As the result is essentially wrongly coding 1 of the causes listed as the UCOD. A better explanation/differentiation between the two types of errors would be helpful.

Competing causes are mutually exclusive conditions reported in different lines of part 1. Multiple causes in a single line are considered a different error category, irrespective of whether the multiple causes in the single line are mutually exclusive or not. If two competing causes are reported on a single line in Part 1, then we classified the error as “Reporting multiple causes in a single line of part 1”. We have clarified this in S2 Text, certification error type 7 “Reporting competing causes in Part 1”.

• Line 123 and footnote 1 : Though the explanation for how SDI is calculated is mentioned, since the authors have mentioned that it is used by Global burden of disease study, a reference to a GBD capstone article/ related GBD reference may be included.

We have added a reference (27) in Line 129.

• Line 136 : Would the decision to include the MCCODs in multiple SDI groups not affect the results? The rationale for including them in multiple SDI groups needs to be mentioned for better clarity. The authors also need to clarify here whether MCCODS with “unspecified neoplasms” error and “poorly defined external causes” error were also out in multiple SDI groups or not.

The decision to include MCCODs in multiple SDI groups is not expected to affect the results because deaths with the same causal sequence would occur in high, medium and low SDI countries. We included the same MCCODs in multiple SDI groups because the process for identifying error-free death certificates was conducted manually and was very time-intensive; hence, this reduced the amount of time needed to identify error-free death certificates. We have now clarified this in Lines 139-141. The “unspecified neoplasms” error and “poorly defined external causes” error were both simulated on MCCODs that were included in multiple SDI groups.

• Fig 1 : SDI categories are not labelled.

The SDI categories in Figure 1 are now labeled.

• Lines 272-273 : Though one of the stated goals was “to monitor trends in the quality of medical certification”, subsequent sections fail to elaborate on this. Though there is the mention of the use of Iris in mainly high SDI countries and the possibility of manual coding causing more errors, there isn’t enough discussion to correlate the study findings and the “quality of medical certification” in other countries.

This sentence should have read “…to enable the monitoring of trends in the quality of medical certification”. We have subsequently modified the existing sentence in Lines 318-320 to be “The weights also allow development of a composite index of certification quality which can be used to monitor trends and differentials in overall certification practices from a database of MCCODs.”

• Lines 279-281 : More details need to be mentioned regarding situations where epidemiological and other public health factors are considered in the UCOD selection. Since these coding rules are mentioned as the main reason for mitigating errors, the situations where they are applied instead of using the condition initiating the causal sequence needs to be elaborated. And if according to the coding rules, some other condition is selected as the UCOD and not the condition initiating the causal sequence, should it still be considered as an error for the purpose of this study?

In mortality coding, whether a causal relationship is considered acceptable or not for mortality coding is established not only on a medical assessment but also on epidemiological and public health considerations. Therefore, a medically acceptable relationship might be listed as unacceptable in the coding instructions because a later step in the sequence is more important from a public health point of view. However, a medically acceptable sequence was never considered as an error for the purposes of this study.

• Line 296 : A brief explanation of Coding rules SP4 and SP5 is needed even though a reference for the same is given (24). This is because of the importance of these rules in understanding why the effect of published “major errors” may not be as high as expected.

Usually, the first cause reported in the lowest used line of part 1 is the underlying cause of death meant by the physician. In an incorrectly reported sequence, the first condition reported in the lowest used line may not explain all the conditions reported above. Rule SP4 allows the coder to search for another possible sequence ending in the first mentioned condition (terminal condition) in part 1 and select the originating cause of that sequence as the tentative starting point. Further, when there is no sequence ending with the terminal condition, the rule SP5 allows the coder to select the terminal condition also as the tentative starting point. We have added this information in a footnote (footnote 2).

Reviewer #6: Dear Authors,

The authors have satisfactorily addressed all the queries by the previous reviewers. This article is very relevant in the present scenario of COVID pandemic. The authors are requested to address the below queries .

We thank the reviewer for their valuable comments. Please find below our responses. All line numbers refer to the document with tracked changes.

Abstract:

• Line 32 and 33: The author needs to clarify whether “ 1592- error free reports” selected were “ either the death occurred in the medical institution or non- institutional death”, since WHO has recommended two types of Forms specifically ie FORM 4 and FORM 4A.

The 1592 error-free MCCODs were US registered deaths that occurred in all settings - i.e. in hospital, at home, in public places, etc. We have now clarified this Lines 124-125.

Discussion:

• Line 328: The author needs to write the type of version and make of “ Iris automated coding software” and the calibration to avoid errors in coding.

Iris version 5.7 was used for coding – this is now mentioned in Line 164.

---

## [Editor Report · Decision Letter 2]

25 Oct 2021

The impact of errors in medical certification on the accuracy of the underlying cause of death

PONE-D-21-14961R2

Dear Dr. Adair,

We’re pleased to inform you that your manuscript has been judged scientifically suitable for publication and will be formally accepted for publication once it meets all outstanding technical requirements.

Kind regards,

Prof. Ritesh G. Menezes, M.B.B.S., M.D., Diplomate N.B.

Academic Editor

PLOS ONE

---

## [Editor Report · Acceptance letter]

29 Oct 2021

PONE-D-21-14961R2 

The impact of errors in medical certification on the accuracy of the underlying cause of death 

Dear Dr. Adair:

I'm pleased to inform you that your manuscript has been deemed suitable for publication in PLOS ONE. Congratulations! Your manuscript is now with our production department. 

Kind regards, 

on behalf of

Prof. Dr. Ritesh G. Menezes 

Academic Editor

PLOS ONE